

# Inconsistent response of soil bacterial and fungal communities in aggregates to litter decomposition during short-term incubation

Jingjing Li[1,2] and  Chao Yang[1]

[1] College of Grassland Science, Qingdao Agricultural University, Qingdao, China
[2] College of Grassland Science and Technology, China Agricultural University, Beijing, China

## ABSTRACT

**Background**. Soil aggregate-size classes and microbial communities within the aggregates are important factors regulating the soil organic carbon (SOC) turnover. However, the response of soil bacterial and fungal communities in aggregates to litter decomposition in different aggregate-size classes is poorly understand.

**Methods**. Soil samples from un-grazed natural grassland were separated into four dry aggregate classes of different sizes (2–4 mm, 1–2 mm, 0.25–1 mm and <0.25 mm). Two types of plant litter (leaf and stem) of *Leymus chinensis* were added to each of the four aggregate class samples. The $CO_2$ release rate, SOC storage and soil microbial communities were measured at the end of the 56-day incubation.

**Results**. The results showed that the 1–2 mm aggregate had the highest bacterial Shannon and $CO_2$ release in CK and leaf addition treatments, and the SOC in the <0.25 mm aggregate was higher than that in the others across the treatments. The relative abundance of Ascomycota was higher in the 2–4 mm and <0.25 mm aggregates than in the 1–2 mm and 0.25–1 mm aggregates in the treatment without litter addition, and the relative abundance of Aphelidiomycota was lower in the 2–4 mm and <0.25 mm aggregates than in the 1–2 mm and 0.25–1 mm aggregates. Also, litter addition increased the relative abundance of Proteobacteria and Bacteroidetes, but decreased the relative abundance of Acidobacteria, Gemmatimonadetes, and Actinobacteria. The relative abundance of Ascomycota and Aphelidiomycota increased by more than 10% following leaf litter addition. The bacterial Shannon index had a significantly positive and direct effect on SOC concentration and $CO_2$ release, while the fungal Shannon index was significantly correlated with SOC concentration. Our results indicate that the soil bacterial diversity contributes positively to both carbon emissions and carbon storage, whereas soil fungal diversity can promote carbon storage and decrease carbon emissions.

Corresponding author
Chao Yang, yc697525@163.com

## INTRODUCTION

Litter decomposition is a key step in the carbon cycle (*Bonan et al., 2013*; *Schmidt et al., 2011*; *Wieder, Bonan & Allison, 2013*), especially in grassland ecosystems, which cover 40%

of the earth's land surface (*Lu et al., 2017*), and possess about 20% of the soil organic carbon (SOC) stocks globally (*Schuman, Janzen & Herrick, 2002*). The majority of studies have shown that litter decomposition usually depends on three main drivers: climate factors (soil moisture and temperature) (*He et al., 2010*; *Riggs et al., 2015*; *Wang, Zeng & Zhong, 2016*; *Zhong et al., 2017*), litter quality (i.e., its chemical composition) (*Hishinuma et al., 2017*; *Zhang et al., 2016*), and decomposer activity (*Jia et al., 2016*; *Keiser & Bradford, 2017*). In addition, there is increasing evidence that soil microbial communities influence litter decomposition rates over and above climate and litter quality controls (*Bradford et al., 2016*; *Schimel & Schaeffer, 2012*). Studies on litter decomposition have focused on the bacterial community composition and have confirmed the involvement of a series of colonizers in the litter decomposition process (*Fanin & Bertrand, 2016*). Soil fungi are also one of the major microbial decomposers of plant litter, being able to decompose the recalcitrant component (*Liang et al., 2017*).

Soil aggregates are the basic units of soil structure (*Bronick & Lal, 2005*) and are generally sub-divided into macro-aggregates (>0.25 mm) and micro-aggregates (<0.25 mm) (*Yang, Liu & Zhang, 2017*). Soil microbial community compositions are non-uniformly distributed in the soil aggregates and may be sensitive to changes in soil environments (*Six et al., 2004*). In general, the distribution of bacteria in soil varies with aggregate size (*Neumann et al., 2013*), and the microbial biomass associated with soil aggregates has been reported to be heterogeneously distributed (*Wang, Li & Zheng, 2017*). Only a few studies have used sequencing analysis to assess the bacterial communities of aggregates (*Gupta & Germida, 2015*). Although our previous study found the distribution of soil bacteria and fungi in aggregates to be inverse (*Yang, Liu & Zhang, 2019*), very little is known regarding their contribution to litter decomposition in different aggregate fractions. In addition, soil microbial processes are regulated by constraints in soil pH, which is considered to be an important factor controlling the balance of fungal to bacterial growth in the soil (*Lauber et al., 2009*; *Rousk et al., 2010*).

We designed a two-factor incubation experiment in the laboratory: one was litter quality (leaf and stem), while the other was soil aggregate size. The SOC mineralization, SOC content, soil pH, and soil microbial community were measured at the end of the incubation period. We hypothesized that (1) the soil bacterial and fungal communities are unevenly distributed in the aggregate fractions, (2) different aggregate sizes respond differently to the quality of the litter addition, and (3) the correlation between soil properties and soil microbial diversities controls litter decomposition.

## MATERIAL AND METHODS

### Material collection and preparation

The soil samples were collected from one type of ungrazed natural grassland located in Guyuan, Hebei Province, China (41°46′N, 115°41′E, elevation 1,380 m) in May of 2018, the initial stage of growing season. This area is a typical temperate zone characterized by a mean annual precipitation of 430 mm and a mean annual temperature of 1.4 °C. The minimum monthly mean air temperature is −18.6 °C in January and the site reaches a

**Table 1** Mean (±standard error (SE), *n* = 3) total carbon (TC), total nitrogen (TN), and carbon to nitrogen ration (C/N) for different soil aggregate size classes and litter types.

| | Soil aggregates | | | | | Litter type | |
| --- | --- | --- | --- | --- | --- | --- | --- |
| | 2–4 mm | 1–2 mm | 0.25–1 mm | <0.25 mm | | Leaf | Stem |
| SOC (g kg$^{-1}$) | 13.27(0.1)ab | 11.87(0.2)b | 5.20(0.2)c | 14.17(0.3)a | TC (g kg$^{-1}$) | 411.96(0.1)c | 424.19(0.1)a |
| TN (g kg$^{-1}$) | 1.67(0.03)bc | 1.73(0.03)b | 1.07(0.07)c | 1.90(0.06)a | TN (g kg$^{-1}$) | 17.20(0.1)a | 14.30(0.1)c |
| C/N ratios | 7.95(0.01)a | 6.86(0.01)b | 4.86(0.02)c | 7.46(0.01)a | C/N ratios | 23.95(0.1)c | 29.66(0.1)a |

Notes.
Different letters in the same row indicate a significant difference at $P < 0.05$ using least significant difference tests.

maximum of 21.1 °C in July. The site has a calcic-orthic Aridisol soil with a loamy-sand texture.

In brief, the top layer (0–15 cm) of the soil (∼200 kg) at one location was collected in plastic bags with a shovel, and was quickly transported to the laboratory by car, upon which the plant roots and leaves were carefully removed by hand and the soil was air-dried. The soil was sieved to separate large macro-aggregates (2–4 mm), macro-aggregates (1–2 mm), meso-aggregates (0.25–1 mm) and micro-aggregates (<0.25 mm) according to *Yang, Liu & Zhang (2017)*. In brief, the undisturbed soil was shaken through four sieves (4, 2, 1 and, 0.25 mm) for 2 min, and the large macro-aggregates (2–4 mm) were collected from the two mm sieve, macro-aggregates (1–2 mm) from the one mm sieve, and meso-aggregates (0.25–1 mm) from the 0.25 mm sieve, and the micro-aggregates (<0.25 mm) were passed through the 0.25 mm sieve. Soil aggregates were stored hermetically at room temperature after until collecting the litter samples.

In September 2018, two types of plant litter (leaf and stem) were obtained from the dominant species (*Leymus chinensis*) in the same location from where the soil samples were collected. The litter was brought to the laboratory, and dried at 65 °C to constant weight. In order to avoid the effects of litter size on decomposition, the plant litter was cut into ca. 1-cm-long sections for the incubation experiment. Some basic characteristics for the soils in Table 1 were cited from our previous studies (*Yang, Li & Zhang, 2019*; *Yang, Liu & Zhang, 2017*; *Yang, Liu & Zhang, 2019*).

**Experimental design and incubation study**
The three replicates of air-dried soil samples (200 g dry weight) of each aggregate size class (2–4, 1–2, 0.25–1, and <0.25 mm) were placed at the bottom of 1,000 mL jars. The jars were new and unused, and we did not sterilize them beforehand because our incubator has the function of ultraviolet sterilization. The two plant litter types (3 g of dry matter) were combined with 200 g of dry soil in the microcosms. Although air drying of soil sample is not representative of the communities that originally existed in the soil, it can represent the difference in the distribution of microbes in our incubation conditions (*Yang, Liu & Zhang, 2019*). No litter addition was used as the control check (CK). There were a total of 72 microcosms (4 aggregate size × 3 litter types × 3 replications × 2 sampling times). The moisture content was adjusted to 30% by weighing each microcosm and adding distilled water, and 30% is the maximum field water capacity of the soil (*Yang, Liu & Zhang, 2019*). Each jar was covered with perforated cling film to reduce humidity loss while allowing

gaseous exchange. The jars were pre-incubated for 3 days at a constant temperature of 25 °C. After the pre incubation period, the jars were maintained for 56 days at 25 °C. During the 56-day incubation, the soil moisture of each microcosm was maintained consistently by weighing each microcosm every week and adding distilled water. After 28 and 56 days of incubation, 36 microcosms were retrieved, respectively. Litter was removed from each microcosm, cleaned with water to remove adhering soil particles, dried (65 °C, 48 h) and weighed.

## Soil aggregate respiration measurements

Soil aggregate respiration was measured at the end of the incubation. In brief, small vials with five mL of 1 M NaOH were placed in the incubation jars to trap $CO_2$ for 24 h. The soil respiration (g $CO_2$- C $g^{-1}$ soil $day^{-1}$) was estimated by titrating two mL NaOH from each trap with 0.1 M HCl after adding two mL 1 M $BaCl_2$ (1:1) and a phenolphthalein indicator using a Digital Burette continuous E (VITLAB, Grossostheim, Germany) according to *Yang et al. (2018)*. At the end of the soil respiration incubation, 10 g of soil sample was collected immediately after the removal of the plant litter and stored at −80 °C for microbiological sequencing. The remaining soil samples were air-dried for SOC and pH assays. SOC concentration was measured using an elemental analyzer (TOC, Elementar, Germany), and soil total nitrogen (TN) was measured using the FOSS Kjeltec 2300 Analyser Unit (FOSS, Hillerød, Sweden). Soil pH was determined after shaking a soil water (1: 2.5 wt/vol) suspension for 30 min.

## Soil DNA extraction and sequencing

Genomic DNA was extracted from each soil aggregate sample using an E.Z.N.A.® stool DNA Kit (Omega Bio-tek, Norcross, GA, USA) according to the manufacturer's instructions. All extracted DNA samples were stored at −20 °C before further analysis. The V3-V4 regions of the bacterial 16S rRNA gene were amplified using universal primers 338F (5′-ACTCCTACGGGAGGCAGCAG-3′) and 806R (5′-GGACTACHVGGGTWTCTAAT-3′) (*Lane et al., 1985*), and the fungal ITS gene was amplified by the ITS1 (5′-CTTGGTCATTTAGAGGAAGTAA-3′) and ITS2 (5′-GCTGCGTTCTTCATCGATGC-3′) primers (*White et al., 1990*). The PCR program was as follows: 5 min initial denaturation at 95 °C; 25 cycles of denaturation at 95 °C (30 s), annealing at 56 °C (30 s), elongation at 72 °C (40 s); and a final extension at 72 °C for 10 min. PCR reactions were performed in triplicate 25 μL mixtures containing 2.5 μL of 10× Pyrobest Buffer, 2 μL of 2.5 Mm dNTPs, 1 μL of each primer (10 Mm), 0.4 U of Pyrobest DNA Polymerase (TaKaRa), and 15 ng of template DNA. The amplicon mixture sequenced on an Illumina HiSeq 2500 platform (Biomaker, Beijing).

## Processing of sequencing data

The extraction of high-quality sequences was first conducted with the QIIME package (Quantitative Insights Into Microbial Ecology) (v1.2.1). The original sequence data were sorted into valid reads after demultiplexing and quality-filtering with the following rules: (i) 300-bp reads were truncated at any site receiving an average quality score of <20 over a 50-bp sliding window, and truncated reads shorter than 50 bp were discarded; (ii) exact

barcode matching, less than two nucleotide mismatches in the primer, and no ambiguous characters in the read; (iii) only overlapping sequences longer than 10 bp were assembled according to their overlapped sequence (*Chen et al., 2018*). The unique sequence set was classified into operational taxonomic units (OTUs) under the threshold of 97% identity using UCLUST.

## Statistical analysis

The Shannon index was calculated as follows (*Begon, Harper & Townsend, 1986*):

$$\text{Shannon} = -\sum \left(\frac{Ni}{N}\right) \ln \left(\frac{Ni}{N}\right);$$

where $N$ is the total OTUs of the sample, $Ni$ is the number of individuals in group $i$.

The SOC, TN, and C/N ratios for the soil aggregates and plant litter were analyzed using a one-way analysis of variance (ANOVA) with a least significant difference (LSD) test at a significance level of $P < 0.05$ using SPSS, version 19.0. Two-way ANOVA was used to test the effects of soil aggregate size and litter type on bacterial Shannon, fungal Shannon, SOC, $CO_2$ release, soil pH and litter mass loss. The structural changes in the soil bacterial and fungal phyla were tested by nonmetric multidimensional scaling (NMDS) based on Bray-Curtis similarity matrices using CANOCO, version 5.0. The relationship between environmental variables (pH and SOC) and bacterial/fungal communities were tested by redundancy analysis (RDA) using CANOCO. For RDA analysis, the significance of the effect of each variable, based on its eigenvalue, was tested using the Monte Carlo Permutation test, and the resulting significance level was determined by the F ratio and *P*-value. Hypothetical relationships among SOC, soil pH, bacterial diversity, fungal diversity, and soil respiration were quantified by structural equation modeling (SEM) using AMOS, version 21.0, and we used the non-significant chi-square ($\chi^2$) test (the model has a good fit when $0 \leq \chi^2 \leq 2$ and $0.05 < P \leq 1.00$) and the root mean square error of approximation (RMSEA, the model has a good fit when $0 \leq \text{RMSEA} \leq 0.05$ and $0.10 < P \leq 1.00$) to test the goodness of the model according to *Yang, Liu & Zhang (2019)*.

## RESULTS

### Microbial diversities and properties of the soil aggregates

The microbial diversities and properties of the soil are listed in Table 2. According to the two-way ANOVA, the 1–2 mm aggregate had the highest bacterial Shannon and $CO_2$ release in CK and leaf addition treatments, and the SOC in the <0.25 mm aggregate was higher than that in the others across the treatments. Soil pH in the 0.25–1 mm aggregate was higher than that in the others across the treatments. In two litter addition treatments, the litter mass loss in the 1–2 mm and 2–4 mm aggregate was significantly higher than that in 0.25–1 mm and <0.25 mm soil aggregate on both day 28 and day 56.

### Response of soil bacteria and fungi to litter addition

The soil bacterial and fungal community structures of the four aggregates were distinct from each other in the NMDS plots in the three litter-addition treatments (Fig. 1). The relative abundances of the dominant bacterial and fungal phyla in the soil aggregates

**Table 2  Diversity of the bacterial and fungal communities and soil properties in each aggregate under litter addition conditions.**

| Litter addition | Sizes | Bacterial Shannon | Fungal Shannon | Soil organic carbon (g kg$^{-1}$) | CO$_2$ release (mg C kg$^{-1}$ day$^{-1}$) | Soil pH | Litter mass loss (%) | |
|---|---|---|---|---|---|---|---|---|
| | | | | | | | Day 28 | Day 56 |
| CK | 2–4 mm | 6.75(0.10) | **4.06(0.16)** | 13.19(0.05) | 61.78(4.9) | 7.93(0.01) | | |
| | 1–2 mm | **6.96(0.03)** | 3.19(0.07) | 11.39(0.10) | **89.47(3.5)** | 8.02(0.01) | | |
| | 0.25–1 mm | 6.83(0.04) | 3.52(0.30) | 5.12(0.01) | 23.28(3.5) | **8.25(0.01)** | | |
| | <0.25 mm | 6.87(0.04) | 4.01(0.04) | **14.10(0.04)** | 43.28(3.5) | 8.02(0.01) | | |
| Leaf | 2–4 mm | 6.77(0.03) | 2.43(0.04) | 14.65(0.10) | 213.95(3.8) | 7.77(0.01) | 29.3(3.2) | 39.4(2.0) |
| | 1–2 mm | **6.87(0.06)** | 2.47(0.05) | 14.66(0.10) | **281.60(0.8)** | 7.82(0.01) | **32.4(1.0)** | **43.3(1.5)** |
| | 0.25–1 mm | 6.72(0.02) | **2.51(0.08)** | 8.60(0.10) | 148.68(1.4) | **7.87(0.01)** | 25.9(2.0) | 33.9(2.1) |
| | <0.25 mm | 6.73(0.03) | 2.45(0.08) | **16.87(0.20)** | 178.68(1.4) | 7.73(0.01) | 16.9(2.1) | 21.9(2.0) |
| Stem | 2–4 mm | 6.81(0.04) | 2.77(0.02) | 14.29(0.12) | 160.42(4.9) | 7.87(0.01) | 26.3(3.0) | 36.3(1.5) |
| | 1–2 mm | 6.82(0.03) | **2.90(0.1)** | 13.61(0.10) | **207.17(5.1)** | 7.91(0.01) | **29.8(2.0)** | **40.8(3.0)** |
| | 0.25–1 mm | 6.65(0.05) | 2.80(0.04) | 9.00(0.50) | 24.75(7.3) | **8.03(0.02)** | 22.1(2.1) | 30.1(3.0) |
| | <0.25 mm | **6.85(0.01)** | 2.47(0.15) | **16.66(0.04)** | 49.75(7.3) | 7.85(0.01) | 10.9(3.0) | 15.9(2.0) |
| Two-way ANOVA | | | | | | | | |
| Litter | | * | *** | *** | *** | *** | ** | ** |
| Sizes | | ** | ** | *** | *** | *** | *** | *** |
| Litter × Sizes | | ns | *** | *** | *** | *** | *** | *** |

**Notes.**

Values are means (±SE) of three measurements.

The highest value among the four aggregate sizes are in bold.

Level of significance:

\*$P < 0.05$.

\*\*$P < 0.01$.

\*\*\*$P < 0.001$.

ns. not significant.

are presented in Fig. 2. *Proteobacteria* and *Acidobacteria* were the main microflora in the soil aggregates in all three treatments, accounting for about 60% of the total abundance (Figs. 2A–2C). The relative abundance of *Ascomycota* was higher in the 2–4 mm and <0.25 mm aggregates than in the 1–2 mm and 0.25–1 mm aggregates in the treatment without leaf and stem addition, and the relative abundance of *Aphelidiomycota* was lower in the 2–4 mm and <0.25 mm aggregates than in the 1–2 mm and 0.25–1 mm aggregates (Fig. 2D). However, leaf and stem addition increased the relative abundance of *Ascomycota* and *Aphelidiomycota* dramatically (Figs. 2E and 2F). The response of soil bacteria to leaf and stem addition was weak, changing less than 10% compare with the treatment lacking litter addition. Leaf and stem addition increased the relative abundance of *Proteobacteria* and *Bacteroidetes*, but decreased the relative abundance of *Acidobacteria*, *Gemmatimonadetes*, and *Actinobacteria* (Figs. 3A, 3B). However, the changes in soil fungi in response to leaf and stem addition were approximately 40% compared with no litter addition treatment, and the relative abundance of *Ascomycota* and *Aphelidiomycota* increased more than 10% following the addition of leaf litter (Figs. 3C, 3D).

## The relationship between soil properties and microbial diversities

RDA biplots were used to assess the physicochemical properties that influenced the abundance of the bacterial and fungal families. Overall, the combination of variables

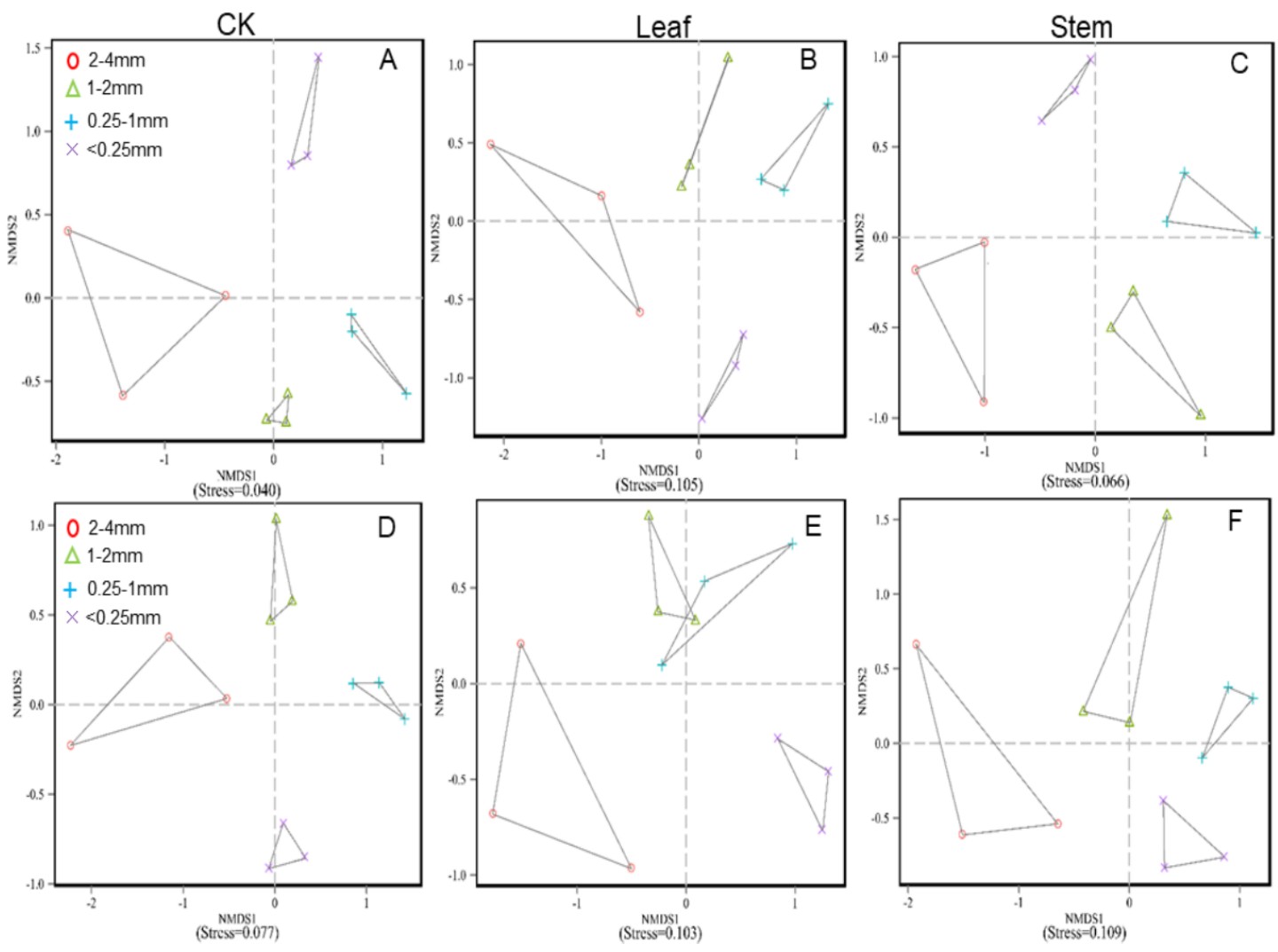

**Figure 1** Nonmetric multidimensional scaling (NMDS) ordinations based on the mean abundance value of the bacterial phyla under no litter addition (A), leaf addition (B), and stem addition (C), and the fungal phyla under no litter addition (D), leaf addition (E), and stem addition (F). Communities are indicated by colored symbols as follows: red circles, 2–4 mm; blue triangles,1–2 mm; green plus signs, 0.25–1 mm, and purple times signs, <0.25 mm.

explained 65.9% and 31.5% of the bacterial (Fig. 4A) and fungal (Fig. 4B) community structure variances, respectively. Significant correlation was found between soil pH and soil bacterial communities ($F = 7.51$, $P = 0.008$), and SOC was also significantly correlated with the bacterial communities ($F = 5.91$, $P = 0.028$). In addition, the fungal communities were significantly correlated with the soil pH ($F = 4.42$, $P = 0.042$) but not by SOC ($F = 2.41$, $P = 0.091$).

The SEM showed a good fit between soil pH, microbial diversity, SOC, and respiration (Fig. 5; $\chi^2 = 0.07$, $P = 0.79$; RMSEA $= 0.00$, $P = 0.80$). The fitted models explained 65% and 59% of the variance in SOC concentrations and $CO_2$ release, respectively (Fig. 5A). Although an interaction was detected, there was no significant correlation between soil

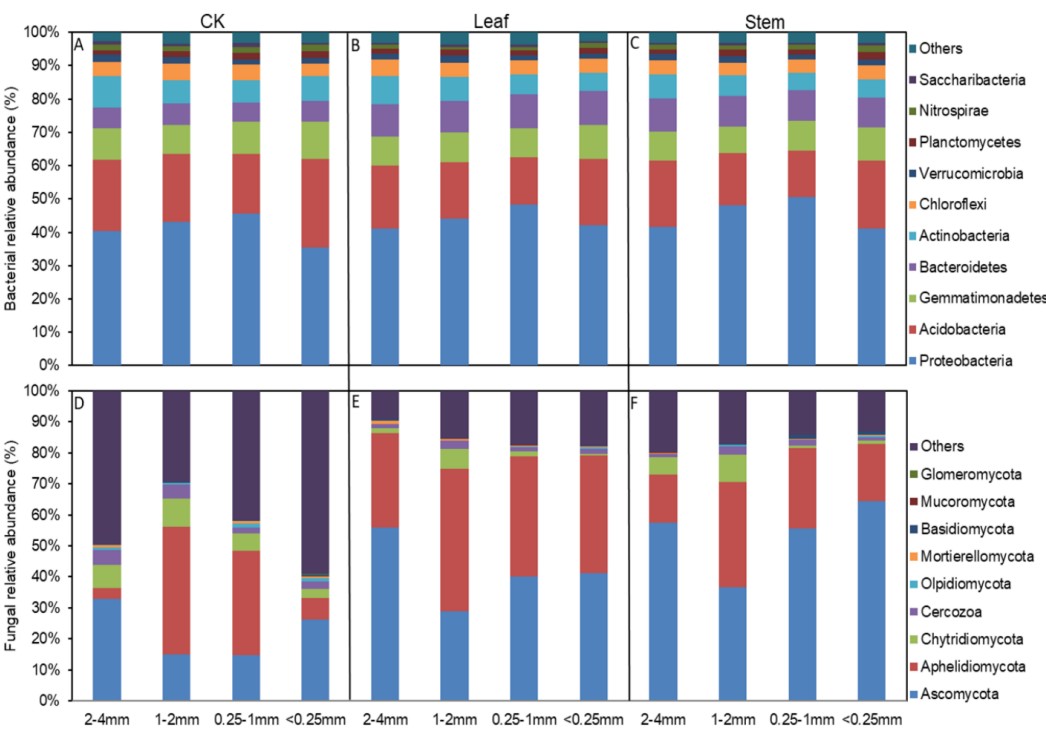

**Figure 2** **The relative abundances of the bacterial (A–C) and fungal phyla (D–F) in different sized aggregates.** The abundances that accounted for <1% of all of the classified sequences belong to Others in the bar graph.

bacterial and fungal diversity with a correlation coefficient ($R^2$) of 0.14. The SEM showed a significant interrelationship between fungal diversity and soil pH ($R^2 = 0.61$, $P < 0.001$). The bacterial Shannon index indicated a significantly positive and direct effect on SOC concentrations and $CO_2$ release ($P < 0.05$), while the fungal Shannon index showed significantly correlated with SOC concentrations ($P < 0.001$). In addition, soil pH was significantly negatively correlated with SOC concentrations and $CO_2$ release ($P < 0.001$). The standardized total effects derived from the SEM revealed that SOC concentrations were mainly driven by soil pH, followed by the fungal and bacterial Shannon index, while $CO_2$ release was mainly driven by soil pH, followed by the bacterial and fungal Shannon index (Fig. 5B)

## DISCUSSION

### Bacterial and fungal diversity in soil aggregates

Soil microbial diversity can better explain respiration than soil microbial biomass (*Yang, Liu & Zhang, 2019*). Furthermore, as macro-aggregates are generally dominated by soil fungi (*Frey, 2005*), our study considered the diversity of both the soil bacteria and soil fungi. The 1–2 mm aggregate possessed both the highest bacterial Shannon and the lowest fungal Shannon under no litter addition, implying that soil bacteria and fungi in the same resources and spaces undergo interspecies interactions at the surface of the macro-aggregates (*Effmert*

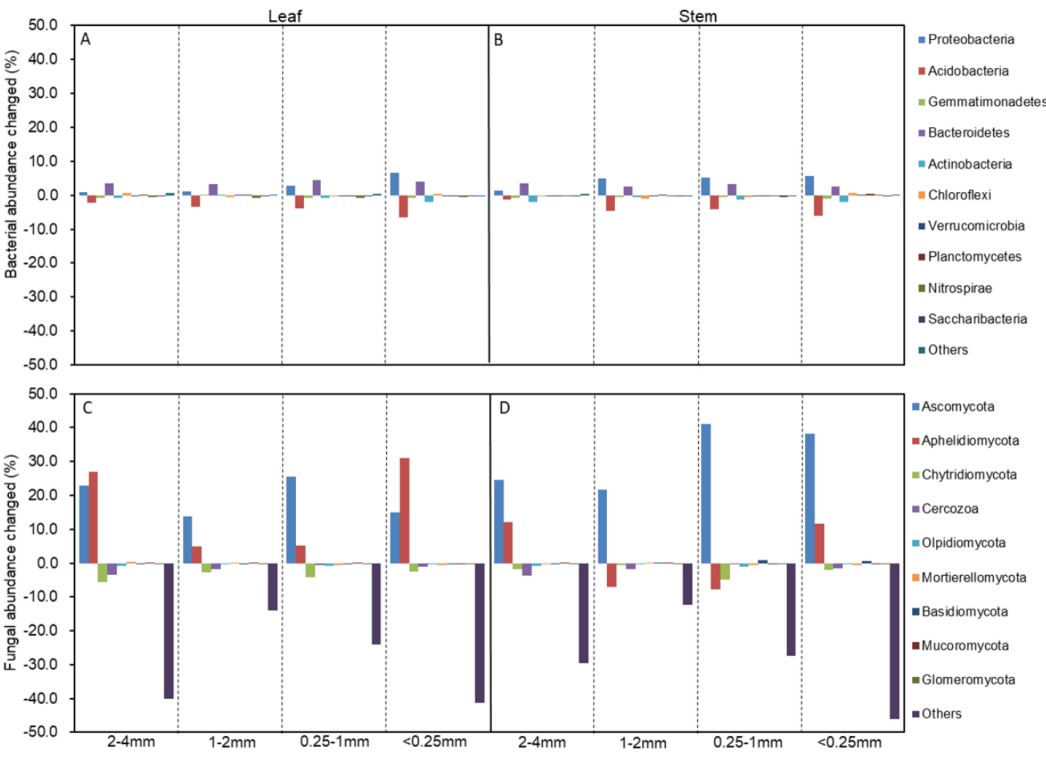

**Figure 3** **The changed abundances of the dominant bacterial (A, B) and fungal phyla (C, D) in different-sized aggregates under litter addition compared with no litter addition.** The abundances that accounted for <1% of all of the classified sequences belong to Others in the bar graph.

*et al., 2012*), and in this aggregate, bacterial diversity is favored. Litter addition eventually resulted in increased respiration; however, the Shannon index of both the bacteria and fungi decreased in the litter-addition treatment compared with no litter addition. *Bamminger et al. (2014)* suggested that litter addition increases microbial biomass and results in a shift in the composition of the soil microbial community, especially for soil fungi, which is consistent with the present study. The relative abundance of individual species will increase dramatically with the addition of litter, particularly that of soil fungi, resulting in a decline in microbial diversity.

## Response of soil bacteria and fungi to litter addition in soil aggregates

Adding leaf litter to the soil has a positive effect on the soil microbial community as a result of the increased carbon and nutrient resources (*Fanin & Bertrand, 2016*). Bacterial communities are characterized by a series of colonizers during litter decomposition, with *Proteobacteria*, *Actinobacteria* and *Bacteroidetes* being the most abundant taxa found over the entire decomposition process (*Purahong et al., 2016*; *Tlaskal, Voriskova & Baldrian, 2016*). Our study demonstrates that *Proteobacteria*, *Actinobacteria*, *Gemmatimonadetes* and *Bacteroidetes* are the four most-important bacterial phyla within different aggregate sizes and litter addition treatments. In addition, *Sun et al. (2017)* found that bacterial abundance
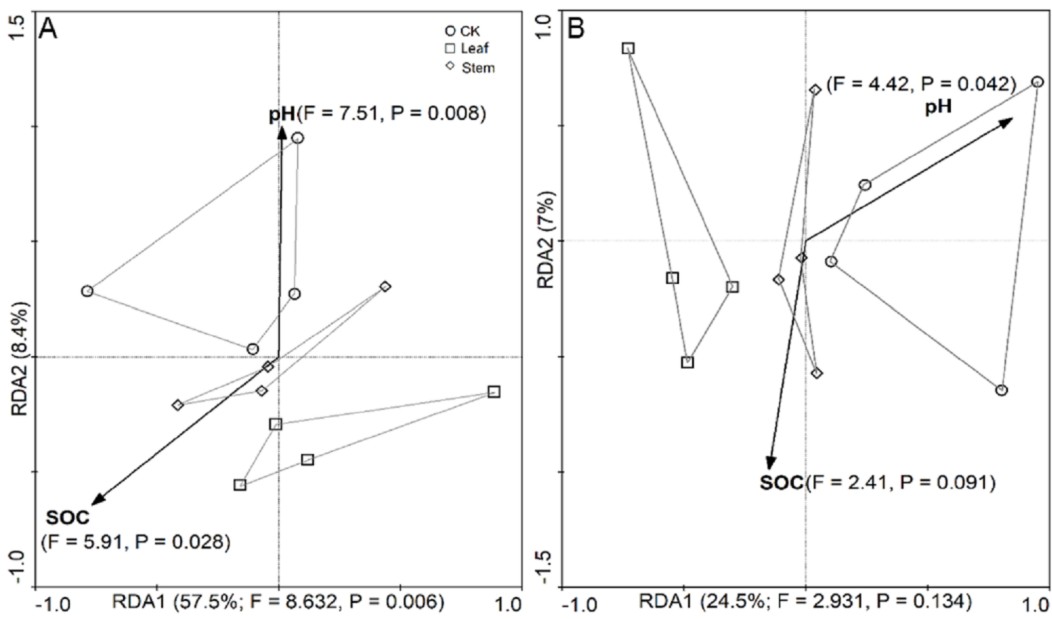

**Figure 4 The redundancy analysis (RDA) showing the impact of soil properties (SOC and pH) on bacterial (A) and fungal (B) community structures.** The significance of the effect of each variable, based on its eigenvalue, was tested using the Monte Carlo Permutation test, and the resulting significance level was determined by the F ratio and *P*-value. Communities are indicated by symbols as follows: circles, no litter addition; squares, leaf addition; diamonds, stem addition.

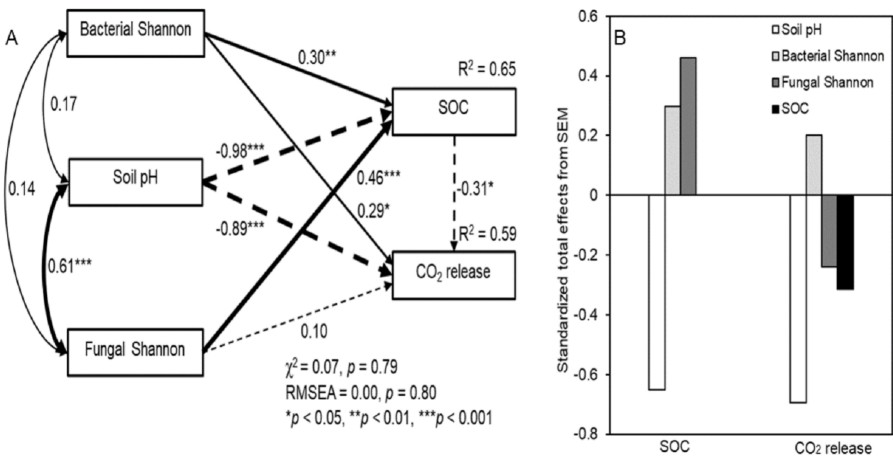

**Figure 5 Structural equation model (SEM) based on the effects of the bacterial and fungal Shannon index on soil pH, SOC, and respiration under litter addition conditions (A), and their standardized total effects (direct plus indirect effects) derived from the structural equation models of SOC and CO2 release (B).** Continuous and dashed arrows indicate positive and negative relationships, respectively. The width of the arrows is proportional to the strength of the path coefficients. R2 indicates the proportion of the variance explained and appears above every response variable in the model.

did not show a significant change following the addition of leaf litter, which corroborates the observations made in the present study. *Liang et al. (2017)* suggest that the addition of residues alters the soil microbial community composition and promotes fungal growth more than bacterial growth, which is in agreement with other studies that found that soil microorganisms respond to altered residue modifications by changing their community composition (*Phillips et al., 2002*) and also supports the observation that saprophytic fungi play a major role in the decomposition of residues (*Meidute, Demoling & Baath, 2008*). Indeed, the fungal relative abundances in our study changed significantly following leaf litter addition, particularly *Ascomycota* and *Aphelidiomycota*. Given that *Aphelidiomycota* are not typical soil fungi, but widely present in soil when litter was added, its role in litter decomposition requires further study.

## Interactions between soil pH, SOC and microbial diversity after litter addition

There is increasing evidence that soil microbial communities influence litter decomposition rates over and above the climate and litter quality controls (*Bradford et al., 2016*; *Schimel & Schaeffer, 2012*). Adding leaf litter to the soil increases the SOC and soil pH (*Sun et al., 2017*), and the soil pH tends to be neutral after adding litter to acid soil. Similarity, in the present study, the soil pH decreased and tended to neutralize following litter addition, implying that litter addition regulates the acid–base balance of the soil. In addition, *Rousk et al. (2010)* suggested that soil pH has a strong influence on the diversity and composition of soil bacterial and fungal communities across a gradient. In contrast, soil microbial diversity also influences soil pH. In the present study, soil fungal diversity decreased after litter addition, which decreased the soil pH. The composition and diversity of the soil fungal and bacterial communities are thus often strongly correlated with soil pH (*Lauber et al., 2009*). In addition, the SEM showed a significant interrelationship between fungal diversity and soil pH, and fungal diversity showed significant correlations with SOC concentrations, especially in the microenvironment of litter addition. The standardized total effects derived from the SEM revealed that SOC concentrations were mainly driven by soil pH, followed by the fungal and bacterial diversity, while $CO_2$ release was mainly driven by soil pH, followed by the bacterial and fungal diversity. Such datasets are valuable in advancing our understanding of the role of different groups of microorganisms in soil and how microbial activities in the soil in response to litter addition contributes to nutrient fluxes in specific soil environments.

## CONCLUSIONS

Our study shows a highly different response of bacteria and fungi in soil aggregates to litter addition. Litter addition increased the relative abundance of *Proteobacteria* and *Bacteroidetes*, but decreased the relative abundance of *Acidobacteria*, *Gemmatimonadetes*, and *Actinobacteria*. The relative abundance of *Ascomycota* was higher in the 2–4 mm and <0.25 mm aggregates than in the 1–2 mm and 0.25–1 mm aggregates in the treatment without litter addition, and the relative abundance of *Aphelidiomycota* was lower in the 2–4 mm and <0.25 mm aggregates than in the 1–2 mm and 0.25–1 mm aggregates. Soil pH

## PeerJ

and SOC were found to be the determining factors shaping the bacterial communities. The bacterial Shannon index had a significantly positive and direct effect on SOC concentration and $CO_2$ release, while the fungal Shannon index showed a significant correlation with SOC concentration. Our results indicate that soil bacterial diversity contributes positively to both carbon emissions and carbon storage, whereas soil fungal diversity can promote carbon storage and decrease carbon emissions.

## ACKNOWLEDGEMENTS

We are very grateful to the workers at the Hebei National Field Research Station of Grassland Science for their help during field work.

### Funding
The authors received no funding for this work.

### Competing Interests
The authors declare there are no competing interests.

### Author Contributions
- Jingjing Li conceived and designed the experiments, performed the experiments, analyzed the data, contributed reagents/materials/analysis tools, prepared figures and/or tables, authored or reviewed drafts of the paper, approved the final draft.
- Chao Yang conceived and designed the experiments, contributed reagents/materials/-analysis tools, prepared figures and/or tables, authored or reviewed drafts of the paper, approved the final draft.

### Data Availability
The sequencing data are available at the Sequence Read Archive (SRA): SRP156109.

### Supplemental Information
Supplemental information for this article can be found online at http://dx.doi.org/10.7717/peerj.8078#supplemental-information.

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

# PeerJ

**Bonan GB, Hartman MD, Parton WJ, Wieder WR. 2013.** Evaluating litter decomposition in earth system models with long-term litterbag experiments: an example using the Community Land Model version 4 (CLM4). *Global Change Biology* **19**:957–974 DOI 10.1111/gcb.12031.

**Bradford MA, Berg B, Maynard DS, Wieder WR, Wood SA. 2016.** Understanding the dominant controls on litter decomposition. *Journal of Ecology* **104**:229–238 DOI 10.1111/1365-2745.12507.

**Bronick CJ, Lal R. 2005.** Soil structure and management: a review. *Geoderma* **124**:3–22 DOI 10.1016/j.geoderma.2004.03.005.

**Chen WQ, Xu R, Wu YT, Chen J, Zhang YJ, Hu TM, Yuan XP, Zhou L, Tan TY, Fan JR. 2018.** Plant diversity is coupled with beta not alpha diversity of soil fungal communities following N enrichment in a semi-arid grassland. *Soil Biology & Biochemistry* **116**:388–398 DOI 10.1016/j.soilbio.2017.10.039.

**Effmert U, Kalderas J, Warnke R, Piechulla B. 2012.** Volatile mediated interactions between bacteria and fungi in the soil. *Journal of Chemical Ecology* **38**:665–703 DOI 10.1007/s10886-012-0135-5.

**Fanin N, Bertrand I. 2016.** Aboveground litter quality is a better predictor than belowground microbial communities when estimating carbon mineralization along a land-use gradient. *Soil Biology & Biochemistry* **94**:48–60 DOI 10.1016/j.soilbio.2015.11.007.

**Frey SD. 2005.** Aggregation-microbial aspects. In: Hillel D, ed. *Encyclopedia of soils in the environment.* Amsterdam: Elsevier Ltd, 22–28 DOI 10.1016/B0-12-348530-4/00130-2.

**Gupta VVSR, Germida JJ. 2015.** Soil aggregation: influence on microbial biomass and implications for biological processes. *Soil Biology & Biochemistry* **80**:A3–A9 DOI 10.1016/j.soilbio.2014.09.002.

**He XB, Lin YH, Han GM, Guo P, Tian XJ. 2010.** The effect of temperature on decomposition of leaf litter from two tropical forests by a microcosm experiment. *European Journal of Soil Biology* **46**:200–207 DOI 10.1016/j.ejsobi.2010.02.001.

**Hishinuma T, Azuma JI, Osono T, Takeda H. 2017.** Litter quality control of decomposition of leaves, twigs, and sapwood by the white-rot fungus Trametes versicolor. *European Journal of Soil Biology* **80**:1–8 DOI 10.1016/j.ejsobi.2017.03.002.

**Jia XQ, He ZH, Weiser MD, Yin T, Akbar S, Kong XS, Tian K, Jia YY, Lin H, Yu MJ, Tian XJ. 2016.** Indoor evidence for the contribution of soil microbes and corresponding environments to the decomposition of Pinus massoniana and Castanopsis sclerophylla litter from Thousand Island Lake. *European Journal of Soil Biology* **77**:44–52 DOI 10.1016/j.ejsobi.2016.10.003.

**Keiser AD, Bradford MA. 2017.** Climate masks decomposer influence in a cross-site litter decomposition study. *Soil Biology & Biochemistry* **107**:180–187 DOI 10.1016/j.soilbio.2016.12.022.

**Lane DJ, Pace B, Olsen GJ, Stahl DA, Sogin ML, Pace NR. 1985.** Rapid determination of 16S ribosomal RNA sequences for phylogenetic analyses. *Proceedings of the National Academy of Sciences of the United States of America* **82**:6955–6959 DOI 10.1073/pnas.82.20.6955.

**Lauber CL, Hamady M, Knight R, Fierer N. 2009.** Pyrosequencing-based assessment of soil pH as a predictor of soil bacterial community structure at the continental scale. *Applied And Environmental Microbiology* **75**:5111–5120 DOI 10.1128/Aem.00335-09.

**Liang X, Yuan J, Yang E, Meng J. 2017.** Responses of soil organic carbon decomposition and microbial community to the addition of plant residues with different C:N ratio. *European Journal of Soil Biology* **82**:50–55 DOI 10.1016/j.ejsobi.2017.08.005.

**Lu WJ, Liu N, Zhang YJ, Zhou JQ, Guo YP, Yang X. 2017.** Impact of vegetation community on litter decomposition: evidence from a reciprocal transplant study with C-13 labeled plant litter. *Soil Biology & Biochemistry* **112**:248–257 DOI 10.1016/j.soilbio.2017.05.014.

**Meidute S, Demoling F, Baath E. 2008.** Antagonistic and synergistic effects of fungal and bacterial growth in soil after adding different carbon and nitrogen sources. *Soil Biology & Biochemistry* **40**:2334–2343 DOI 10.1016/j.soilbio.2008.05.011.

**Neumann D, Heuer A, Hemkemeyer M, Martens R, Tebbe CC. 2013.** Response of microbial communities to long-term fertilization depends on their microhabitat. *Fems Microbiology Ecology* **86**:71–84 DOI 10.1111/1574-6941.12092.

**Phillips RL, Zak DR, Holmes WE, White DC. 2002.** Microbial community composition and function beneath temperate trees exposed to elevated atmospheric carbon dioxide and ozone. *Oecologia* **131**:236–244 DOI 10.1007/s00442-002-0868-x.

**Purahong W, Wubet T, Lentendu G, Schloter M, Pecyna MJ, Kapturska D, Hofrichter M, Kruger D, Buscot F. 2016.** Life in leaf litter: novel insights into community dynamics of bacteria and fungi during litter decomposition. *Molecular Ecology* **25**:4059–4074 DOI 10.1111/mec.13739.

**Riggs CE, Hobbie SE, Cavender-Bares J, Savage JA, Wei XJ. 2015.** Contrasting effects of plant species traits and moisture on the decomposition of multiple litter fractions. *Oecologia* **179**:573–584 DOI 10.1007/s00442-015-3352-0.

**Rousk J, Baath E, Brookes PC, Lauber CL, Lozupone C, Caporaso JG, Knight R, Fierer N. 2010.** Soil bacterial and fungal communities across a pH gradient in an arable soil. *Isme Journal* **4**:1340–1351 DOI 10.1038/ismej.2010.58.

**Schimel JP, Schaeffer SM. 2012.** Microbial control over carbon cycling in soil. *Frontiers in Microbiology* **3**:Article 348 DOI 10.3389/Fmicb.2012.00348.

**Schmidt MWI, Torn MS, Abiven S, Dittmar T, Guggenberger G, Janssens IA, Kleber M, Kogel-Knabner I, Lehmann J, Manning DAC, Nannipieri P, Rasse DP, Weiner S, Trumbore SE. 2011.** Persistence of soil organic matter as an ecosystem property. *Nature* **478**:49–56 DOI 10.1038/nature10386.

**Schuman GE, Janzen HH, Herrick JE. 2002.** Soil carbon dynamics and potential carbon sequestration by rangelands. *Environmental Pollution* **116**:391–396 DOI 10.1016/S0269-7491(01)00215-9.

**Six J, Bossuyt H, Degryze S, Denef K. 2004.** A history of research on the link between (micro)aggregates, soil biota, and soil organic matter dynamics. *Soil & Tillage Research* **79**:7–31 DOI 10.1016/j.still.2004.03.008.

**Sun H, Wang QX, Liu N, Li L, Zhang CG, Liu ZB, Zhang YY. 2017.** Effects of different leaf litters on the physicochemical properties and bacterial communities in Panax ginseng-growing soil. *Applied Soil Ecology* **111**:17–24 DOI 10.1016/j.apsoil.2016.11.008.

**Tlaskal V, Voriskova J, Baldrian P. 2016.** Bacterial succession on decomposing leaf litter exhibits a specific occurrence pattern of cellulolytic taxa and potential decomposers of fungal mycelia. *Fems Microbiology Ecology* **92**:Article fiw177 DOI 10.1093/femsec/fiw177.

**Wang QK, Zeng ZQ, Zhong MC. 2016.** Soil moisture alters the response of soil organic carbon mineralization to litter addition. *Ecosystems* **19**:450–460 DOI 10.1007/s10021-015-9941-2.

**Wang SQ, Li TX, Zheng ZC. 2017.** Distribution of microbial biomass and activity within soil aggregates as affected by tea plantation age. *Catena* **153**:1–8 DOI 10.1016/j.catena.2017.01.029.

**White TJ, Bruns T, Lee S, Taylor J. 1990.** Amplification and direct sequencing of fungal ribosomal RNA genes for phylogenetics. In: Innis M, Gelfand D, Sninsky J, White TS, eds. *PCR protocols.* San Diego: Academic Press DOI 10.1016/b978-0-12–8.50042-1.

**Wieder WR, Bonan GB, Allison SD. 2013.** Global soil carbon projections are improved by modelling microbial processes. *Nature Climate Change* **3**:909–912 DOI 10.1038/Nclimate1951.

**Yang C, Li JJ, Zhang YJ. 2019.** Soil aggregates indirectly influence litter carbon storage and release through soil pH in the highly alkaline soils of north China. *PeerJ* **7**:e7949 DOI 10.7717/peerj.7949.

**Yang C, Liu N, Zhang YJ. 2017.** Effects of aggregates size and glucose addition on soil organic carbon mineralization and Q(10) values under wide temperature change conditions. *European Journal of Soil Biology* **80**:77–84 DOI 10.1016/j.ejsobi.2017.04.002.

**Yang C, Liu N, Zhang YJ. 2019.** Soil aggregates regulate the impact of soil bacterial and fungal communities on soil respiration. *Geoderma* **337**:444–452 DOI 10.1016/j.geoderma.2018.10.002.

**Yang C, Zhang YJ, Rong YP, Bei YX, Wei YQ, Liu N. 2018.** Temporal variation of Q(10) values in response to changes in soil physiochemical properties caused by fairy rings. *European Journal of Soil Biology* **86**:42–48 DOI 10.1016/j.ejsobi.2018.03.001.

**Zhang WD, Chao L, Yang QP, Wang QK, Fang YT, Wang SL. 2016.** Litter quality mediated nitrogen effect on plant litter decomposition regardless of soil fauna presence. *Ecology* **97**:2834–2843 DOI 10.1002/ecy.1515.

**Zhong YQW, Yan WM, Wang RW, Shangguan ZP. 2017.** Differential responses of litter decomposition to nutrient addition and soil water availability with long-term vegetation recovery. *Biology and Fertility of Soils* **53**:939–949 DOI 10.1007/s00374-017-1242-9.