# Peer review of "Inconsistent response of soil bacterial and fungal communities in aggregates to litter decomposition during short-term incubation"

_PeerJ, doi:10.7717/peerj.8078_

## Round 0.1 · original submission · Major Revisions

Based on three peer reviews, I am recommending major revisions to your manuscript. Overall, the paper has the potential to contribute to improved understanding of the role of aggregate size in litter decomposition. However, the hypotheses do not align well with the experimental design as described by reviewer #3. Also, this same reviewer recommends improvement to the discussion section so that it is more closely linked to the specific results of your study. Reviewer #1 challenges the validity of obtaining DNA sequences only at the end of the 56-day study. Please better justify or qualify this approach. Finally, reviewer #3 requests that you are clearer about your results that reflect biodiversity and not abundance. Please clarify throughout, especially in the discussion. If you decide to revise the paper, please respond in detail to the comments of each of the reviewers.

Reviewer 1 ·

Basic reporting

In this manuscript, Li and Yang describe a report in regards to the inconsistent response of soil bacterial and fungal communities in aggregates to litter decomposition. However, not only the designed experimental approaches raised many concerns regarding its scientific sounds, scarce data obtained from this study support the conclusions in this manuscript. Below, I have some comments in regards to additional experiments and modified approach that would help authors improve the quality of their manuscript in order to meet a general standard for publishing as an original research article.

While the authors invested their effort in bioinformatics analyses, which obtained from DNA sequencing of specific regions (16S rRNA and ITS1-2), all of the data showing in this manuscript were observed in only one time point at the end of the incubation period (56-day after incubation) without a statement of specific or logical reasons. In my opinion, these data are insufficient to reflect a robust, convincible, and reliable interpretation. Rather, the author should perform a comprehensive comparison with at least three-time point: 0-, 28- (sometime in the mid-process), 56-day after incubation. This designed strategy would hopefully provide insights into dynamic responses of soil bacterial and fungal communities in aggregates to litter decomposition.

There are many concerns with regards to material collection and preparation. While the collection of top layer (0–15 cm) of the soil was stated, there were lacking detailed description regarding the representative of sample, including how many samples were collected, any specific season, ect.

For the preparation of materials, how was the contamination of un-indigenous microorganisms controlled during transportation and sample preparation? Importantly, the 1000 mL jars were utilized for incubation, how was the aeration managed, given that this is normally a major factor regulating the balance of fungal and bacterial growth in the soil.

The authors should clarify why was the Leymus chinensis chose as plant litter (leaf and stem) in this study.

Experimental design

As above

Validity of the findings

No comment

Reviewer 2 ·

Basic reporting

The manuscript is well written. Clear English is used throughout for the easy understanding of readers. Sufficient background and citations are also provided.

Experimental design

The experimental design is well constructed.

Validity of the findings

Please see my review attached.

Annotated reviews are not available for download in order to protect the identity of reviewers who chose to remain anonymous.

Reviewer 3 ·

Basic reporting

1. Basic Reporting
The English is largely very good, but there are some minor corrections outlined below.
The raw data appear to all be present and correct. The only omission is that Sheet 2 of the .xlsx should include the sample names as present in the SRA, so that the reads can be matched to the correct sample.
Specific comments on abstract/ introduction section:
Line 8: please provide an academic/professional e-mail address if possible
Line 13: please remove the comma after ‘grassland’
Line 15: there were four aggregate classes, not three
Lines 17-23: the summary of the results makes no mention of the different aggregate classes, and whether the observed patterns differed across them. Please add in some information on this.
Line 28: avoid repetition of the word ‘factors’, e.g. “…three main drivers: climate factors…”
Line 42: I’m not sure what you mean by ‘converse’. Do you mean there was an inverse relationship?
Lines 46-51: the hypotheses section needs some work. The whole set up of the experiment is designed to test whether different aggregate sizes respond differently to litter addition, and whether the quality of the litter has any effect on the relationship. Neither of these is mentioned in the hypotheses. In addition, please justify hypothesis 2. The layout of the hypotheses, results section and discussion section should mirror one another in order to present a clear story.

Experimental design

2. Experimental design
The stated hypotheses need some work (see above). In particular, the main research questions are implicit rather than being fully stated.
Specific comments on methods section:
Line 57: what area or weight of soil was removed?
Line 63: how was the soil stored between May and September?
Lines 66, 70: was the plant material dried, or did you obtain dry weight from a subsample?
Line 69: you mention in your previous paper (Yang, 2019) that air-drying soil can alter the microbial communities. If so, it would be good to a) discuss that in the present paper (where community analysis is central) and b) if this is experimentally demonstrated, back it up with a reference to the literature.
Line 71: why was the control designated CK?
Line 81: 2 ml NaOH from (not of) each trap
Lines 83-85: did you measure weight loss in the plant material as a indicator of decomposition? If so, this would be very interesting data to add.
Line 94: when citing primers, please refer to the paper that originally described them
Line 97: I assume you mean µL (microlitre) not mL (millilitre)! Please check carefully throughout the rest of the paper that you have referred to the correct unit.
Lines 99-100: this sentence doesn’t make sense. Was it sequenced on a HiSeq in the USA or a MiSeq in China?
Lines 104-108: Step i) why a 300 bp cutoff, when the target region was aprrox. 450 bp? Step ii) this suggests that exact barcode matches were removed - I assume you mean barcode errors were removed. Step ii) it’s not clear what this means.
Line 112: as for the primers, please cite the source of these equations or a reference work (eg Begon, Harper and Townsend) rather than a paper that simply used them.
Line 117: please give these abbreviations in full at the first mention. Also, do you mean a two-way ANOVA, seeing that you give two predictor variables?
Line 118: were the LSD P-values adjusted for multiple comparisons?
Line 118: please capitalise P (here and throughout)
Line 120: you use both Shannon and Chao1, which capture different aspects of diversity, but at no point do you discuss how they differ or what they tell you. If you are going to include more than one index, please include this information and discuss it in the discussion section.

Validity of the findings

3. Validity of the Findings
The conclusions section makes no mention of aggregate size classes, which is strange given that this was the main component of the experiment. In general, statements in the discussion need to be better justified and checked carefully to see if they can really be derived from these results. For example, lines 216-217 “Soil fungi diversity plays a more important role in litter decomposition than soil bacterial diversity”. This is probably true, but I cannot see how it could be derived from the results of the current study.
Always keep in mind that you measured diversity, not abundance, of bacterial and fungal communities. This means that you only have a measure of change within each community, not between them. It is therefore not possible to make any statements about a shift between bacterial vs fungal dominance (e.g. line 189).
Specific comments on results and discussion:


Results:
The first section of the results comes across as rather confused. It needs reworking to take the reader through each ANOVA model in turn and tell a coherent story.
There is no need to report main effects where there is a significant interaction, as the two factors depend on each other. In the first paragraph, you mention leaf but not stem addition, and sometimes refer to litter addition without specifying whether this is leaf, stem or both treatments lumped together. Be very careful about statements suggesting causation, e.g. line 160-161 – you can’t actually tell whether the pH and SOC influenced the bacterial communities or vice versa, as the relationship is only correlative.
The section on SEM could do to have a bit more information to make it clear for those unfamiliar with the method (like me). In particular, the meaning of lines 166-167 is unclear.
Line 134: there is no need for the word “on”

Fig 1: if the data were analysed together, do treatments also separate from each other?
Fig. 4 would benefit from more information in the legend.

Discussion:
I’m puzzled by the appearance of Aphelidiomycota in your results. This is not a decomposer taxon, but aquatic parasites associated with algae (Karpov et al. 2014. Front Microbiol 5: 112). I think you should look carefully into the identification of these OTUs and how come they are in your data. Also, the Cercozoa are not fungi but protists. They presumably were co-amplified by the fungal primers, and really should be excluded from the analysis as these primers do not deliberately target protists.
The model shown in Fig 5 seems to be one of the most important parts of your manuscript, yet it isn’t really mentioned in the discussion.
Lines 177-178: what do you mean by the proportion of bacteria? The bacteria-fungus ratio? I feel that the first few lines of the discussion would be better placed in the introduction, as they don’t directly relate to interpreting your findings.
Lines 183-184: I don’t see how you can imply competition from these results.
Line 197: bacterial phyla
Lines 198-199: be careful! You didn’t measure abundance, so in what way do Sun’s findings corroborate yours?
Line 204: again, the fungal communities didn’t increase, but alter in composition

Additional comments

4. General comments
This is a well-designed study, which explores an often-overlooked aspect of soil diversity: the physical heterogeneity of soil, and the different microbial communities associated with different aggregate size classes. However, the presentation and interpretation of the results needs considerable work in order to be clear and robust.

---

## Round 0.2 · Minor Revisions

I have received comments from two reviewers and request that you make some additional minor revisions to your paper. Reviewer #3 notes that the paper is much improved over the previous version, but request some additional minor changes. Reviewer #1 raises some sharper criticisms which, while valid, should not prevent eventual publication of this work. Regarding reviewer #1 comments, point #1 that the experiment be re-done with additional measurements at 28 days is not valid in my opinion as you have stated that this is a 56-day experiment and you did add additional data on litter loss at 28 days. Regarding the second point about soil sampling, I ask that to the extent possible you include information in the paper about how many soil samples, collection in one location or not, and representativeness of the soil sampling. Finally, regarding point #3, please add information that the jars were new and unused but that you did not sterilize them beforehand. I do not view this criticism as being especially strong in this case because the abundance or microorganisms in the soil samples would be expected to completely overwhelm and dilute any contribution from the small number that may have been present in the jars.

Reviewer 1 ·

Basic reporting

The authors have revised and added some additional information as the request in the review report. However, the revised manuscript did not address major concerns that would improve its quality and meet a scientific sound. Again, all the experimental procedures from sample harvesting to sample preparation and experiment design were insufficient to obtain robust, reliable, and repeatable results. Importantly, there were no alternative approaches to validate the achieved data represented in the manuscript. For publishing as an original research article, this study should be thoroughly re-conducted with additional sets of well-designed experiments, thereby hopefully obtaining scientific-sounded evidence for the questions the author asked. Below are comments based on the authors' responses:

1. While the authors invested their effort in bioinformatics analyses, which obtained from DNA sequencing of specific regions (16S rRNA and ITS1-2), all of the data showing in this manuscript were observed in only one time point at the end of the incubation period (56-day after incubation) without a statement of specific or logical reasons. In my opinion, these data are insufficient to reflect a robust, convincible, and reliable interpretation. Rather, the author should perform a comprehensive comparison with at least three-time point: 0-, 28- (sometime in the mid-process), 56-day after incubation. This designed strategy would hopefully provide insights into dynamic responses of soil bacterial and fungal communities in aggregates to litter decomposition.

Response: Thank you for the kind suggestion. We are sorry that we only measured soil microbial structure at the end of the incubation period (56-day after incubation). We have supplemented the data of litter mass loss after 28 and 56 days of incubation in Table 2, which helps us to understand the temporal dynamics of litter decomposition, and to make the paper more logical, we mention that this experiment is a short-term incubation experiment.

Comment: I am not convincing by this response. In this case, a comprehensive comparison with at least three-time point: 0-, 28- (sometime in the mid-process), and 56-day after incubation should be performed for all of the CO2 release rate, SOC storage and soil microbial communities. Otherwise, it seems to be inadequate to prove the role of soil bacterial and fungal communities in aggregates to litter decomposition. Additionally, I do not agree with the statement “during” in line 17, because it was only at the end of incubation process.


2. There are many concerns with regards to material collection and preparation. While the collection of top layer (0–15 cm) of the soil was stated, there were lacking detailed description regarding the representative of sample, including how many samples were collected, any specific season, ect.

Response: Thank you for the kind suggestion. The soil samples were collected from a natural grassland located in Guyuan, Hebei Province, China (41°46′ N, 115°41′ E, elevation 1380 m) in May of 2018, the initial stage of growing season. The site has a calcic-orthic Aridisol soil with a loamy-sand texture.

In brief, the top layer (0–15 cm) of the soil (~200 kg) was collected in plastic bags with a shovel, and was quickly transported to the laboratory by car, upon which the plant roots and leaves were carefully removed by hand and the soil was air-dried.

Comment: This response did not answer exactly how many samples were collected. Additionally, based on this response, ~200 kg of soil were collected at only one location? Any specific proposal for this soil collection? Why designed square-sample model and representative approaches were not applied in this field work?


3. For the preparation of materials, how was the contamination of un-indigenous microorganisms controlled during transportation and sample preparation? Importantly, the 1000 mL jars were utilized for incubation, how was the aeration managed, given that this is normally a major factor regulating the balance of fungal and bacterial growth in the soil.

Response: Soil aggregates were stored hermetically at room temperature after until collecting the litter samples. The 1000 mL jars are new and unused, so that the contamination of un-indigenous microorganisms can be avoided as much as possible. Also, each jar was covered with perforated cling film, which can not only keep the air permeability but also avoid the contamination of un-indigenous microorganisms.

Comment: In a typical microbial investigation, “The 1000 mL jars are new and unused” is inadequate to support the idea “the contamination of un-indigenous microorganisms can be avoided”. Therefore, a sterile condition should be applied. Importantly, how can sterile condition be assured for the section “Experimental design and incubation study” in line 79-93?

Experimental design

As above

Validity of the findings

As above

Reviewer 3 ·

Basic reporting

See below

Experimental design

See below

Validity of the findings

See below

Additional comments

I would like to commend the authors for their improvements to the manuscript, which have greatly improved it. There are still a few points I would like addressed prior to publication, which are listed below:

Line 17: “control” not CK as the abbreviation hasn’t yet been defined
74: laboratory, and dried
76: cited, not sited
120-122: still doesn’t full make sense. How about this? “(ii) exact barcode matching, less than two nucleotide mismatches in the primer, and no ambiguous characters in the read”
125: which database?
178 &179: significantly correlated with
202: still implies competitive dominance, which can’t be assessed from the data. I would suggest “in this aggregate, bacterial diversity is favoured”
216-217: This is still misleading – it currently tells the reader that both you and Sun et al. found that litter addition did not change the bacterial abundance, ie the total number of individual bacterial cells. Actually, what you both found was that litter addition did not change the bacterial relative abundance, ie community composition. This says nothing about total abundance, because if one taxon increases another will decrease relative to it. Conversely, the number of cells could double between samples, but if the taxonomic composition was unchanged you would detect no difference.
221-22: see previous point – you can’t say that fungal relative abundances increased, because what did they increase in relation to? You can say that fungal relative abundances changed, and that the relative abundance of specific taxa increased.
223: I concur that Aphelidiomycota are fungi, but based on their typical ecology their appearance is very surprising. I would therefore like to see some discussion of this point, at least to explain to the reader that they are not typical soil fungi.

---

## Round 0.3 · accepted · Accept

Thank you for addressing these remaining minor concerns regarding your manuscript. I am satisfied with the responses and changes you have made to the paper and recommend that your manuscript be accepted for publication.